# *Coxiella burnetii* in Dogs and Cats from Portugal: Serological and Molecular Analysis

**DOI:** 10.3390/pathogens11121525

**Published:** 2022-12-13

**Authors:** Sofia Anastácio, Samuel Anjos, Suzi Neves, Tiago Neves, Pedro Esteves, Hélder Craveiro, Bruno Madeira, Maria dos Anjos Pires, Sérgio Sousa, Gabriela da Silva, Hugo Vilhena

**Affiliations:** 1Vasco da Gama Research Centre (CIVG), Department of Veterinary Sciences, Vasco da Gama University School, Avenida José R. Sousa Fernandes 197 Lordemão, 3020-210 Coimbra, Portugal; 2Center of Neurosciences and Cell Biology, Health Science Campus, Azinhaga de Santa Comba, 3000-548 Coimbra, Portugal; 3Faculty of Pharmacy, University of Coimbra, Azinhaga de Santa Comba, 3000-548 Coimbra, Portugal; 4Baixo Vouga Veterinary Hospital (HVBV), EN 1, 355, 3750-742 Águeda, Portugal; 5University Veterinary Hospital of Coimbra (HVUC), Lordemão, 3020-210 Coimbra, Portugal; 6Animalstaff, R. Quinta da Várzea, Santa Clara, 3030-092 Coimbra, Portugal; 7Animal and Veterinary Research Centre (CECAV), University of Trás-os-Montes and Alto Douro (UTAD), Quinta de Prados, 5000-801 Vila Real, Portugal; 8Department of Veterinary Sciences, University of Trás-os-Montes and Alto Douro (UTAD), Quinta de Prados, 5000-801 Vila Real, Portugal; 9Associate Laboratory for Animal and Veterinary Sciences-AL4AnimalS, 1300-477 Lisbon, Portugal

**Keywords:** Q fever, pets, PCR, ELISA, zoonosis

## Abstract

Dogs and cats are potential sources of infection for some zoonotic diseases such as Q fever, caused by *Coxiella burnetii*, a multiple host pathogen. Q fever outbreaks in dogs and cats have been related with parturition and abortion events, and ticks have a potential role in the transmission of this pathogen. This study aimed to screen for *C. burnetii* in dogs and cats, and in ticks collected from infested animals. An observational descriptive study was conducted in Portugal at two time points nine years apart, 2012 and 2021. Sera obtained from dogs and cats (total n = 294) were tested for *C. burnetii* antibodies using a commercial ELISA adapted for multi-species detection. *C. burnetii* DNA was screened by qPCR assay targeting IS1111 in uterine samples and in ticks. A decrease in the exposure to *C. burnetii* was observed in cats from 17.2% (95% CI: 5.8–35.8%) in 2012 to 0.0% in 2021, and in dogs from 12.6% (95% CI: 7.7–19.0%) in 2012 to 1.7% (95% CI: 0.3–9.1%) in 2021 (*p* < 0.05). Overall, and despite differences in the samples, rural habitat seems to favour the exposure to *C. burnetii*. The DNA of *C. burnetii* was not detected in ticks. The low seropositivity observed in 2021 and the absence of *C. burnetii* DNA in the tested samples, suggest that dogs and cats from Portugal are not often exposed to the pathogen. Nevertheless, the monitoring of *C. burnetii* infection in companion animals is an important tool to prevent human outbreaks, considering the zoonotic potential for owners and veterinarians contacting infected animals, mainly dogs and cats from rural areas which often come into contact with livestock.

## 1. Introduction

*Coxiella burnetii*, a small Gram-negative intracellular bacterium, is the causative agent of Q fever, a zoonotic infection distributed worldwide. The host range of *C. burnetii* includes mammals, birds, reptiles, and arthropods [1]. Domestic ruminants are recognized as the main sources of human infection and the shedding occurs mostly at the time of parturition, when the heavily infected placenta results in aerosolization of *C. burnetii* [2]. However, human Q fever outbreaks have also been documented from the contact with dogs and cats [3,4,5,6,7]. Pets can potentially be infected by inhalation, tick bites, consumption of placentas or milk from infected ruminants [8]. The finding of *C. burnetii* in vaginal and uterine samples of healthy cats [9,10,11,12] and dogs [11,12,13,14] suggests a potential zoonotic risk for humans. In cats, experimental infection with *C. burnetii* can cause fever, anorexia and lethargy; however, in the field, the infection remains asymptomatic and frequently undiagnosed [15]. In infected parturient dogs, early death of the pups has been reported [3].

Few reports indicate that ticks, ectoparasites of mammals including pets, can act as vectors of *C. burnetii* [16,17,18]. Ticks acquire *C. burnetii* during a blood meal on infected animals. In ticks, transstadial and transovarian transmission may occur. Transmission from ticks to vertebrates might occur during the next blood meal or by aerogenic spread of dried tick faecal excretions [17,19,20]. Over than 40 species of ticks have been found to carry *C. burnetii*, and eventually they may serve as indicators of infection in nature [16,17,18].

This study aimed to evaluate the exposure to *C. burnetii* infection in dogs and cats from central Portugal over a period of nine years apart, and to detect the presence of *C. burnetii* DNA in uterine samples of bitches and queens, and in ticks collected from companion animals.

## 2. Materials and Methods

### 2.1. Study Design

An observational descriptive study was conducted in the central region of Portugal at two time points nine years apart, 2012 and 2021. For sample size calculations, an expected prevalence of 4.8% was considered [21] following the method described by [22], considering a desired absolute precision of 10% and a 95% confidence interval. The calculation was performed using the software WinEpiscope version 2.0. Samples were collected by a convenience sampling strategy in dogs and cats. In 2012, the following inclusion criteria were considered: males and females older than six months and attending to veterinary medical care centres or housed at a municipal kennel. In 2021, the inclusion criteria: females older than six months and attending to veterinary medical care centres for ovariohysterectomy procedures.

### 2.2. Dogs and Cats Attending to Veterinary Medical Care Centres

To evaluate the exposure to *C. burnetii,* the surplus plasma or serum samples of whole blood collected into EDTA tubes or into tubes without anticoagulant during the routine procedures of veterinary examination were kept, after receiving written consent of the owners (2012 n = 99; 2021 n = 107), and stored at −20 °C until analysis. 

In 2012, in dogs and cats selected for blood sampling, ticks were collected (n = 97) and preserved in 70% alcohol. All ticks were identified by stereomicroscopy using a standard morphological key [23] and the data was recorded, including the developmental stage (i.e., larval, nymph, adult) and the gender (i.e., male and female). Ticks were used to detect the presence of *C. burnetii* DNA.

In 2021, reproductive tissue, and/or endometrial swabs were collected from females submitted to ovariohysterectomy (n = 107). The confidentiality of the patients and the well-being of the animals was ensured during the whole study. Furthermore, none of the sampling procedures were performed with the strict purpose of this study.

### 2.3. Dogs and Cats from Municipal Kennels

In 2012, blood samples were collected into tubes without anticoagulant (n = 81) from anesthetized animals undergoing to specific surgical procedures. This procedure was carried out under approval of Portuguese National Authority for Animal Health (no.C.12.014.UDER).

### 2.4. Dogs and Cats Details

In both periods of sampling, individual information was registered in a questionnaire containing characterization variables such as the gender (i.e., pets sampled in 2012), breed, age, habitat, and exposure to wildlife or to other domestic animals.

### 2.5. Antibody Testing

Sera and plasma were tested in duplicate using a commercial indirect ELISA kit, including phase I and phase II *C. burnetii* antigens obtained from infected ruminants and adapted for multi-species (ID Screen Q Fever Indirect Multispecies^®^, IDVet) following the manufacturer’s instructions. Furthermore, the S/P% was calculated through the optical density values at 450 nm obtained in each sample. Thus, results were categorized as negative when S/P% ≤ 40%, doubtful when S/P% > 40% and ≤50%, positive when S/P% > 50%.

### 2.6. Molecular Testing

To obtain DNA for PCR testing the commercial Qiamp DNA Mini Kit (Qiagen^®^, Isaza, Portugal) was used following to manufacturer’s instructions.

### 2.7. DNA Extraction from Ticks

Ticks were washed three times in 1× phosphate-buffered saline, rinsed with distilled water, and dried on sterile filter paper prior to DNA extraction [24]. Then, they were crushed individually and aseptically, using surgical scalpel blades in sterile Petri dishes and 25 mg of the obtained homogenized material was used for DNA extraction.

### 2.8. DNA Extraction from Reproductive Tissue and Endometrial Swabs

Uterine pieces and/or endometrial swabs were pooled in groups of seven to ten samples. In uterine pieces, 25 mg of tissue was collected aseptically, using scalpel blades, and homogenized in a sterile Petri dish. Subsequently, 25 mg of the pooled homogenate was used for DNA isolation. Endometrial swabs were suspended into 1000 µL of sterile phosphate-buffered saline (PBS), then 200 µL of each swab was placed into an Eppendorf^®^ tube and vortexed. Finally, 200 µL of the whole suspension was used for DNA extraction. 

### 2.9. Real-Time PCR

A real-time PCR assay targeting an insertion sequence used for detecting *C. burnetii*, the IS1111, was conducted using the commercial Taq Vet™ *Coxiella burnetii* Real-time PCR kit (LSI^®^, Lissieu, France) on a CFX-96 thermocycler (Bio-Rad^®^, Amadora, Portugal) to determine the threshold cycle (Ct) values, following the manufacturer instructions. This commercial assay was developed as a duplex PCR to detect *C. burnetii* DNA in ruminant samples and a ruminant reference gene (GAPDH) used as an internal positive control. Because this commercial kit was used in non-ruminant samples, a duplicate assay was performed to control the presence of Taq polymerase inhibitors. So, in one assay 5 µL of species DNA was used (i.e., dog or cat), and in the other assay 2.5 µL of the species DNA plus 2.5 µL of ruminant DNA (*C. burnetii* negative) was used. This procedure aimed to screen the presence of false negative results due to Taq polymerase inhibitors in samples from other species rather than ruminants.

### 2.10. Statistical Analyses

For statistical analyses, a descriptive analysis was performed, and a simple logistic regression analysis was performed to explore associations between individual factors and response variables. Confidence limits for the proportions were estimated with 95% confidence intervals (CI) assuming a binominal exact distribution (EpiInfo v7.2.5.0; Center for Disease Control and Prevention, Atlanta, GA, USA).

## 3. Results

### 3.1. Antibody Testing

A total of 287 serum or plasma samples were tested from dogs (n = 211) and cats (n = 76). The obtained results in each serosurvey and per species are described in Table 1.

In 2012, the mean age of the tested animals was estimated in 63.6 months (range: 6 to 192 months, SD = 60.8 months) in cats and 71 months (range: 6 to 168 months, SD = 51 months) in dogs. Male gender was slightly predominant in both species: 55.2% in cats (95% CI: 36–73%) and 62.9% in dogs (95% CI: 53.7–71.3%); as well as the rural habitat: 58.6% of cats (95% CI: 39.1–75.9%) and 54.9% of dogs (95% CI: 46.0–63.6%) lived in a rural area.

An exposure to *C. burnetii* was evidenced in five cats (17.2%; 95% CI: 5.8 to 35.8%). The proportion of positive results was slightly higher in female cats (10.3%, 95% CI: 2.7–28.5%) and in cats younger than 24 months (10.3%, 95% CI: 2.7 28.5%). All the positive cats lived in rural areas. Among them, 80% (95% CI: 29.9 98.9%) referred the exposure to wildlife and 20% (95% CI: 1.1–7.0%) cohabited with other domestic animal species.

In dogs, an exposure to *C. burnetii* was evidenced in 19 animals (12.6%; 95% CI: 7.7–19.0%). Despite the lack of data in some animals, the proportion of positive results was equal in both genders as well as in rural and urban areas. A higher proportion of positive results was found in animals older than 24 months (5.3%; 95% CI: 2.7–10.1%), in animals with an owner (8%; 95% CI: 4.4–13.8%), and in dogs not cohabiting with domestic animals (4.0%; 95% CI: 1.6–8.8%) but exposed to wildlife (4.0%; 95% CI: 1.6–8.8%).

In 2021, the mean age was estimated in 14.3 months (range: 6 to 108 months, SD = 19.9 months) in cats, and 52 months (range: 6 to 180 months, SD = 46.6 months) in dogs. All the samples were taken from females and a predominance of an urban habitat was observed in dogs (n = 56; 93.3%; 95%CI: 83.0–97.8%) and in cats (n = 47; 100%). Regarding the contact with other animal species, despite the lack of information in several animals, only 12 cats (25.5%; 95% CI: 14.4–40.6%) and 11 dogs (18.3%,95% CI: 9.9–30.9%) had contact with other animal species. The contact with domestic ruminants was not mentioned in any of the studied cats.

No exposure to *C. burnetii* was found in cats. However, it was evidenced in one dog (1.7%; 95%, CI: 0.1–10.1%). It corresponded to a companion 6-year-old Chihuahua living in a rural area, apparently with no contact with other animal species.

Overall, a decrease on the proportion of antibody positive samples was observed from 2012 to 2021. The difference of these results was statistically significant in dogs (*p* < 0.05).

### 3.2. Molecular Testing

#### 3.2.1. Ticks

Table 2 summarizes the data about tick identification and PCR results, in 2012. A total of 91 hard ticks comprising 88 adults (96.7%) and 3 nymphs (3.3%) were collected. Among those, 77 (84.6%) were collected from dogs and 14 (15.4%) from cats. In the adult forms, females were predominant (61; 69.3%) and most of them (40/61; 65.6%) were engorged. The most common species in both dogs and cats was *Rhipicephalus sanguineus*, followed by *Ixodes ricinus*. *Dermacentor reticulatus* was only identified in one dog.

*C. burnetii* DNA was not detected in ticks. In duplicate assays, the internal reference gene was amplified. The results for external positive controls were positive and for external negative controls were negative. Thus, PCR reagents were validated as well as the absence of inhibitors of the Taq polymerase in DNA samples.

#### 3.2.2. Uterine Tissue and/or Endometrial Swabs

Among uterine tissue and endometrial swabs collected from queens and bitches, in 2021, all the samples were negative for the presence of *C. burnetii* DNA. In duplicate assays, the internal reference gene was amplified. The results for external positive controls were positive and for external negative controls were negative. Thus, PCR reagents were validated as well as the absence of inhibitors of the Taq polymerase in DNA samples.

## 4. Discussion

This study was conducted at two time points nine years apart. In the first serosurvey, conducted in 2012, the proportion of positive results was 17.2% and 12.6% in cats and dogs, respectively. In the second serosurvey, in 2021, no positive results were obtained in cats and 1.7% of dogs evidenced the presence of *C. burnetii* antibodies. In the same region, at the same time of the first survey (2012), a serosurvey was conducted in small ruminants and a global individual seroprevalence of 9.6% was estimated [25]. However, a significant increase in seroprevalence was observed in sheep from the central region of Portugal a few years later [26]. The results obtained in the second survey (2021) seem to discard the scenario observed in small ruminants. Nevertheless, a careful analysis must be done since one gap in this study is the convenience sampling triggering bias in the obtained results. Furthermore, it should not be neglected that the second survey occurred during the COVID-19 pandemics which certainly affected the sampling and the results since due to the mandatory confinement, pet dogs and cats principally from urban areas reduced their contact with other animals.

Studies conducted in cats reported an exposure to *C. burnetii* of 14.2% in Japan [27], 9% in California, United States [28], 13% in Zimbabwe and 2% in South Africa [29]. A serosurvey conducted in Canada-Nova Scotia, where an outbreak of human Q fever was reported [5], revealed a seroprevalence of 6% in cats [30]. Interestingly, in a similar study performed in the region of Ontario, no seropositive results were found in cats [31] revealing that the patterns of infection may differ geographically, during time or depending on sampling strategy. In fact, a higher seroprevalence (61.5%) was obtained in cats from the United Kingdom but the sampling strategy included cats with owners, living outdoors with hunting habits which might have biased the results by the exposure to wildlife [32]. In the presented study, in the second serosurvey (2021) cats living in an urban area were predominant (96%). This finding differs from the first serosurvey (2012) in which more than 50% of the cats lived in rural areas and among them, 80% (95% CI: 29.9–98.9) were exposed to wildlife (data not shown). Unfortunately, in 2021, the registration of these data failed hampering comparisons. Overall, despite the constraints related with the sampling, it seems that the higher exposure to *C. burnetii* observed in cats in 2012 might be explained by the higher percentage of animals originated from rural areas. In these areas, the likelihood of contact with potential sources of infection such as wild animals or livestock is higher comparing with urban areas [5,32]. In Japan, a seropositive rate of 41.7% was found in stray cats, being higher than the seropositivity in domestic cats (14.2%) [27]. These findings demonstrate that the feline environment influences the exposure to *C. burnetii*.

In dogs, similarly to cats, a significant decrease on the proportion of antibody positive animals was observed from 2012 (12.6%) to 2021 (1.7%). In 1995, in a serosurvey conducted in Portugal 4.8% of dogs evidenced an exposure to *C. burnetii* [21]. Previous surveys of *C. burnetii* in dogs reported an exposure of 26.1% in Australia [14], 66% in California [28], 59.3% in Bulgaria [33], 12% in Croatia [34], 9.8% in France, 11.6% in Senegal [35], 1% in Italy [36] and in Canada no antibodies were detected in dogs [30]. Differences on seroprevalence may occur during time and between geographic locations. Nevertheless, the comparison of data of published studies must be done carefully due to differences in study design, sampling approach and methods applied.

Interestingly, in 2012, a higher antibody positivity (8.0%) was found in dogs with owner comparing with those from the municipal kennel (4.6%) (data not shown). These findings are discordant with those obtained in Montenegro where a higher positivity was obtained in shelter dogs [37]. Additionally, in the United States, California, a higher positivity was observed in stray dogs than in dogs with owner [28]. The herein presented results are somewhat unexpected considering that most of the animals kept at the municipal shelter were captured from the streets and thus presented a higher chance of exposure to *C. burnetii* by the contact with wildlife [1]. Moreover, and despite the rural origin of the unique positive result in 2021, in 2012 a similar positivity was obtained in dogs from rural (5.3%) and from urban (5.3%) areas (data not shown). These findings are also unanticipated since a higher positivity would be expected in rural areas, due to contact with livestock and wildlife favouring the exposure to *C. burnetii* [1]. In urban areas Q fever outbreaks might be related to windborne spreading of *C. burnetii* [38] but questions about the potential sources of infection in urban areas remain unclear.

*C. burnetii* shows a high tropism towards uterus and infective bacteria can be found in high concentrations in the birth products from infected females [1]. Thus, in 2021, uterine tissue and endometrial swabs were collected during ovariohysterectomy procedures to detect DNA of *C. burnetii* by PCR testing. However, no positive PCR results were obtained. Similar results were obtained in Australia [14]. These results might be due to the absence or low concentration of the pathogen in the tested samples. At the time of sample collection most of the females were not pregnant and it is known that the number of bacteria increases drastically during pregnancy due to its massive replication in placental tissues [12,39]. 

In Portugal, the climate and ecological conditions are favourable for the development of several species of ticks and 24 species have been identified so far [40]. *C. burnetii* DNA was not detected in any of the three species of ticks collected from dogs and cats, in 2012. Similar results were obtained in the Netherlands [41], Poland [42], Switzerland [43], Sweeden [44] and Japan [45]. However, positive results were obtained in Belarus (1%) [46], Slovakia (3%) [47], Poland (15.9%) [20], Italy (22%) [41] and in pools of ticks from Kenya (50%) [48]. In these studies, *C. burnetii* was found in the genus *Rhipicephalus*, *Ixodes*, *Dermacentor* and *Haemaphisalis* collected in dogs or in the environment [9,41,45,46,48]. These results should be carefully examined because of limited number of ticks analysed. Thus, we must consider that if these results were confirmed by extensive studies using ticks collected from animals and from the environment, the risk of acquiring *C. burnetii* from ticks would be considered negligible in Portugal. Notwithstanding, the identification of a hotspot of *C. burnetii* in ticks at a regional level in Belarus [46] and the high prevalence of *C. burnetii* in ticks collected in a public park in Rome, suggest the potential role of ticks in the maintenance of infection in some regions [41]. In this context, monitoring of vectors should be performed on a regular basis.

In this study, it was not possible to establish an association between the presence of *C. burnetii* in pets and ticks taken from infested animals. However, ticks cannot be discarded as potential vectors and thus more investigation is needed to really clarify the role of ticks in the transmission of *C. burnetii*, namely at the species level and their habitat.

The exposure to *C. burnetii* was confirmed in both cats and dogs. Overall, cats seem to be more predisposed to be exposed to *C. burnetii*, which might be related to their living habits since indoor cats having frequently free access to the backyard where they express their hunting instinct, and possibly being exposed to potential infected wildlife prey. In dogs, the sources of infection are not well described. 

Globally, dogs and cats seem to play a minor role in the transmission of *C. burnetii*. However, a few outbreaks related to pets’ transmission have been reported in which affected individuals confirmed the contact with parturient animals testing positive to *C. burnetii*. Furthermore, stillbirth and perinatal mortality have been described in dogs and cats and linked with human Q fever outbreaks [3,4,6,7,49,50].

## 5. Conclusions

Our findings suggest that, in Portugal, dogs and cats are exposed to *C. burnetii*. However, the exposure rate seems to change over time. *C. burnetii* DNA was not detected in ticks or in uterine tissue or in endometrial swabs. Thus, it is suggested that neither dogs nor cats nor ticks seem to play a major role in the transmission of *C. burnetii* or be a source for human infections for this agent in Portugal.

## Figures and Tables

**Table 1 pathogens-11-01525-t001:** Results from the serosurvey conducted, in 2012 and in 2021, in dogs and in cats.

Study Period	Cats			Dogs		
	Number (n)	Number (%) of Positives	^a^ CI 95%	Number (n)	Number (%) of Positives	^a^ CI 95%
2012	29	5 (17.2)	5.8–35.8	151	19 (12.6)	7.7–19.0
2021	47	0 (0)	n.a. ^b^	60	1 (1.7)	0.3–9.1
Total	76	5 (6.6)	2.5–15.3	211	20 (9.5)	6.0–14.5

^a^ confidence interval, ^b^ not applicable.

**Table 2 pathogens-11-01525-t002:** Characterisation of the ticks identified in dogs and cats, in 2012, and PCR results.

Tick Species	Stage/Sex					Nº Ticks	Nº Hosts	PCR Result
	Engorged Female	Non-Engorged Female	Male	Nymph	Larva			
Dogs								
*R. sanguineus*	34	16	23	1	0	74	14	Negative
*I. ricinus*	0	0	1	1	0	2	2	Negative
*D. reticulatus*	0	0	1	0	0	1	1	Negative
Cats								
*R. sanguineus*	6	3	1	0	0	10	6	Negative
*I. ricinus*	0	2	1	1	0	4	3	Negative

## Data Availability

Not applicable.

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
