# Peer review of "Coxiella burnetii in Dogs and Cats from Portugal: Serological and Molecular Analysis"

_pathogens, 2022, doi:10.3390/pathogens11121525_

Round 1

Reviewer 1 Report

This is an interesting study into a neglected zoonotic disease. Generally this is a well written and clear paper. 

From a reader's perspective it would be useful for an explanation in the  of why 2012 and 2021 were chosen. It is mentioned that this was opportunistic sampling however given the differences in the outcomes (positives in 2012 but not 2021) it something that is not really mentioned. Could it be the sampling  was insufficient in 2021? This needs to be discussed in more detail

Minor:

Discussion: Please check the word spacing and change 'tutor' to owner throughout.

Reviewer 2 Report

The manuscript entitled “Coxiella burnetii in dogs and cats from Portugal: serological and molecular analyses” by Anastacio et al., evaluates the exposure of C. burnetii infection in dogs and cats in parallel with detection of C. burnetii’s DNA in uterine samples and in ticks collected from pets.

The MS needs substantial editing. The materials and methods, the results part respectively the discussion must be revised. The materials and methods are written in a manner, it is heard to follow. Until these fundamental issues are not revised, I cannot recommend the acceptance of the MS.

L62: ‘and the infection is maintained in the ticks’ – in ticks it’s not causing infection. Revise the sentence.

Materials and methods

It is described in a chaotic way. Divide the ‘Study design and sampling approach’ l: 72-102 into 3 sections:

Study design – l 72-80

Sampling – dividing the kennels and cats/dogs into different subtitle: e.g. 1. Sampling, 1.1. Kennels (describe what samples were collected from kennels), 1.2. Dogs/cats – details

L95: uterine samples of what?

L101: information regarding the gender of which animal?

L116: how the homogenate was obtained?

Molecular analysis: should be divided also to multiple subtitle, molecular analysis of ticks, blood etc. l113-122 -  it’s not a description of molecular analysis, this section must be moved to the sampling subtitle of each animal category

No confirmatory test was used for ELISA? E.g. serum neutralization test

Table 1: dogs and cats… in which samples of these animals?

Table 2: what does it mean missing? Unidentified? Must be included a total round at the end of each category: male-female-unindentifed-total, etc

L150, 154: results repeated - presents similar results as in table 2

What about uterine tissues results? It must be dedicated a separate subtitle, it heard to find the results if these.

Discussion

L231: pervious study – when was performed?

Discuss what could be the reason of the 0% prevalence in ticks.

Reviewer 3 Report

The work described by Anastacio is a seroprevalence study for Coxiella burnetii in dogs and cats in Portugal in 2012 and 2021. The number of samples used is quite low. Also, not all information is available for all animals. In fact, given the lack of information about the animals sampled in 2021, it is statistically impossible to highlight any difference between the two groups other than that of total seroprevalence. The authors also attempted to identify the pathogen from ticks and vaginal swabs, again with a limited number of samples and a commercial amplification kit. Molecular analysis resulted in no identification. The manuscript contains numerous spelling errors, and a number of sentences need to be rewritten because they are difficult to understand. The discussion would have to be almost entirely rewritten, focusing on the data obtained and comparing it to those obtained in other studies, explaining any differences and limitations, and avoiding simply listing the results obtained in other studies. Given the limited data obtained, among other things, on a small number of animals, I would suggest that the authors consider converting the article into a communication.

Title: Please review your title spacing.

Abstract:

Line 26: The authors wrote: “caused by the multiple host pathogen Coxiella burnetii”. Please, rephrase it “multiple host pathogen”.

Line 26-28: “Some Q fever outbreaks have been related with parturition and abortion events in dogs and in cats. Additionally, ticks are considered C. burnetii vectors in wild and domestic cycles.” Could be changed in: “Q fever outbreaks in dogs and cats have been related with parturition and abortion events, as well as ticks have a potential role in the transmission of this pathogen” or similar.

Line 36: The authors wrote: ”Ticks (n=91) were identified as Rhipicephalus sanguineus, Ixodes ricinus and Dermacentor reticulatus, the former being the most common”. This is not relevant for the abstract section.

Line 63: Please, change “transtadically and transovarially transmission” in “transstadial and transovarian transmission”.

Line 87: Is it relevant to know that the analysis has been conducted on "surplus serum"?

Line 98: The same for Line 87.

Line 99: Please remove this sentence, it is explained later: “Ticks, uterine samples, and endometrial swabs were used to detect C. burnetii DNA”.

Line 125: double brackets

Line 126: The author wrote:” To confirm that no inhibitions of Taq Polymerase occurred samples were tested in duplicate. In one sample 5 µL of species DNA was used and in the other sample, a mixture of 2.5 µL of the species DNA was associated with 2.5 µL of C. burnetii negative ruminant DNA.”. This is unclear, but it did not appear to be relevant to the manuscript. 

Tabella 1 e 2: Authors should add a column to the table with the number of positives (and not just the percentage of positives) for that row, so that the table is more intuitive.

Tabella 3: the information for the year 2021 is incomplete, if not completely absent. It is difficult to make a comparison. This table doesn't make much sense, it could be dropped. Just as in the results and discussions, one can only comment on the final prevalence figure between several years, and not on the trend of the various risk factors.

The authors should mention why all the swamp and the ticks resulted in negative results in their analysis. Small sampling? References about shedding in pets?

Discussion:

192-205: This part is not important, please dele it.

Line 224: The authors wrote: “Overall, the observations of 2012 can be explained by the predatory activity 224 of cats in wildlife, living close to prey animals or even to livestock”. I can’t understand this difference. Why were the animals predatory in 2012 but not in 2021? 

Line 231: Space.

Line 231-233: Please rephrase the following sentence: “Aprevious serosurvey conducted in Portugal reported the presence of antibodies in 4.8% of tested dogs [20], but the different sensibility and specificity of methods applied and differences on sampling strategy should not be neglected”.

Line 241: What the authors intend for “in dogs with tutor”?

Line 242-243: Please rephrase this sentence “which is opposite to that found in California, where a higher positivity was observed in stray dogs than in dogs with owner”.

Line 246: The authors wrote “These findings are somewhat unexpected”. Why? This sentence could be coupled with the following.

Line 254: “In females C. burnetii shows a tropism to uterine tissues” rephrase.

Line 257-260: Why was this aspect not taken into account during sampling?

Line 261-274: Does not add any discussion to the manuscript, you can delete.

Line 277-284: The same. How could the description of ticks gathered be useful for the discussion section? 

Line 292-293: Moderate the tone of the sentence. You only tested a small sample, in other countries the positivity was still there. You could start the sentence with “If the results were to be confirmed by extensive studies etc…”.

Line 303: “Prone” is not the right term.

Line 306: The authors wrote: “ the potential sources of infection need to be clarified.”. Why? How?

Line 307-308: “Globally, pets seem to show a less important role in the transmission of C. burnetii, 307 but they can also be a reservoir and source of transmission to humans.” This sentence seems a contradiction.

Line 309: “ All the reports have in common the fact that the infected people were near the parturient animal, also that the newborns died shortly after delivery or were already dead and that the animal also tested positive to C. burnetii”. This sentence needs to be reformulated absolutely.

Line 315: “ seemsto” (space).

Round 2

Reviewer 2 Report

I have no more comments to the authors.

Reviewer 3 Report

In this revision, the authors have satisfactorily addressed most of my previous comments and questions, and I would like to thank them.